# Relationship between the Transport Coefficients of Polar Substances and Entropy

**DOI:** 10.3390/e22010013

**Published:** 2019-12-20

**Authors:** Ivan Anashkin, Sergey Dyakonov, German Dyakonov

**Affiliations:** Chemical Process Engineering Department, Kazan National Research Technological University, Kazan 420063, Russia

**Keywords:** equation of state, entropy, diffusion coefficient, viscosity coefficient, thermal conductivity coefficient

## Abstract

An expression is proposed that relates the transport properties of polar substances (diffusion coefficient, viscosity coefficient, and thermal conductivity coefficient) with entropy. To calculate the entropy, an equation of state with a good description of the properties in a wide region of the state is used. Comparison of calculations based on the proposed expressions with experimental data showed good agreement. A deviation exceeding 20% is observed only in the region near the critical point as well as at high pressures.

## 1. Introduction

Transport coefficients (diffusion *D*, viscosity η, and thermal conductivity λ coefficients) are necessary for almost any algorithm related to designing the chemical technology processes. Theoretical calculation of the transport coefficients is based on kinetic equations of non-equilibrium distribution functions of the Boltzmann equation type. However, to formulate these expressions in a form acceptable for calculation is not feasible. In this regard, several approaches were elaborated to calculate the transport properties of the substances.

The first approach is molecular dynamic simulation. The trajectories of molecules are calculated during simulation. The intermolecular interaction potential must be set for the substances under investigation. The values of diffusion coefficients can be calculated through the autocorrelation function by the Kubo Green relation or by the mean square deviation of the molecule position [1]. Some approaches also exist for calculating the viscosity and thermal conductivity coefficients. These methods provide acceptable accuracy of calculation, but require high computational efforts, and accordingly, much time for calculation. State-of-the-art computer technology does not allow them to be used in chemical and technological processes design. However, the molecular methods remain the only method that allow studying the process at the micro level. In addition, they are often used to test various theories or semi-empirical methods.

In practice, for the calculation of transport properties, semi-empirical and empirical expression are used. Usually the use of such expressions is possible only in a limited region of thermodynamic state, but the expressions reproduce the experimental data with high accuracy. This approach cannot be used if it is required to develop a prototype of a new process, for which the properties have not yet been measured. Now, the Chemical Abstracts Service (CAS) database [2] contains more than 156 million organic and inorganic substances and this amount keeps growing at a high rate. Hence, it is impossible to rely on each experiment pursuing the purposes of obtaining the data on physicochemical properties. The experimental data are not always available or have been measured only in a narrow range of thermodynamic states. Therefore, the approaches to be developed for the calculation of physicochemical properties of the substances are based only on information of the molecule structure.

The properties of substances are determined by the interaction between molecules. This allows us to conclude that the various properties must be interconnected. In this regard, in thermodynamics for temperature, pressure, enthalpy, etc. rigorous relations were obtained that linked them to each other. Binding relationships were also proposed for the transport properties of substances [3,4,5,6], but they are not rigorous and are based on various models and assumptions. The idea of the global isomorphism of properties was widely developed [7,8,9,10,11]. The idea of isomorphism is based on the proposition that in dimensionless variables, the properties of different substances have the same, or similar, behavior. Many successful applications of using excess entropy to scale transport properties were shown in [12].

In [13,14,15], various properties of the substances were investigated in the studies, and it was demonstrated that determining a relation between the properties of two points on the phase diagram is possible. The relationship between these states can be described in a quasi-chemical approximation. The change in property from one point to another can be described by a transformation potential. For the transformation potential, one can use any of the following thermodynamic potentials: Gibbs energy *G*, Helmholtz energy *f*, and entropy *S*, for example.

Entropy was often used to describe transport properties. A confirmation of the relationship was obtained on the basis of simplified expression for macroscopic [16] and microscopic properties [17,18]. Many empirical relationships were proposed as well, the performance of which was tested on a variety of experimental data for real substances [19] and numerical simulations for the model fluids [20,21,22]. Entropy was used to describe the transport properties of amorphous materials [23].

Similar to this work, an approach was used to describe viscosity of nonpolar molecules and hydrocarbons [24,25,26,27]. In the studies, ideal gas has been used as a reference point for comparison, and the viscosity coefficient was described by the expression: $\eta =\eta_{0}Ae^{B\frac{S-S_{0}}{k_{B}}}$, where *A* and *B* are the parameters dependent on the substance, and the index «0» corresponds to the ideal gas state. However, this approach was not applied to polar substances and our calculations for water demonstrated that the expression proposed in [24,25,26,27] could not be applied within the entire range of states.

Thus, in the present research work empirical expressions establishing relationships between transport properties of polar substances and entropy are proposed. Thermodynamic functions can be derived from the equation of state. In the present paper, we show only the possibility of applying the proposed method for calculating transport properties. Therefore, in this work, we used the equations of state obtained from a large number of experimental data.

An ideal theoretical method for calculating the physicochemical properties of pure substances and its mixtures should be based only on the chemical structure of the molecule, which determines the strength of the interaction between the molecules. In our work, one of such stages of a calculation is proposed—the connection of transport properties with equations of state. The equation of state can be obtained from the parameters of intermolecular interaction by molecular simulation or using some approximations [28]. In turn, the potential of intermolecular interaction can be calculated using the ab initio methods [29,30,31,32].

## 2. Equation of State and Transport Properties

The calculations were made for water, ammonia, methanol, and ethanol. These substances were chosen due to the fact that high-precision equations for thermodynamic and transport properties were already published for these polar substances before. Aside from the above argumentation, we also proceeded from practical application. Water is one of the most common substances used in the chemical industry. It is used as a reagent, heat carrier, solvent, etc. Moreover, the intermolecular interaction of water is specific, and the formation of hydrogen bonds causes a number of features (high critical temperature, maximum density at 4 ${}^{\circ }$C). To describe the intermolecular interaction of water, many models [33] were offered, which differ not only in the value of parameters but also in the number and arrangement of interaction centers that again complicates the implementation of molecular simulation.

The intermolecular interaction of polar substances was described by the Stockmayer potential [34]:(1)$$\varphi \left(r\right)=4\varepsilon \left({\left(\frac{\sigma }{r}\right)}^{12}-{\left(\frac{\sigma }{r}\right)}^{6}+\delta {\left(\frac{\sigma }{r}\right)}^{3}\right),$$
where ε, σ, and δ are parameters characterizing the depth of the potential well, the size of the molecule, and the dipole moment, respectively.

The used parameters of intermolecular interaction are presented in Table 1.

To eliminate the error associated with the determination of thermodynamic properties, we used a high-accuracy equation of state [37,38,39,40]. This equation generalizes the extensive number of experimental works and deviation from the experimental values in the wide temperature and pressure ranges is commonly less than 5%. The equation of state expresses the specific Helmholtz free energy. This equation is divided into two parts, an ideal gas part and a residual part:(2)$$\frac{f(\tau ,\theta )}{RT}=\phi^{0}\left(\tau ,\theta \right)+\phi^{e}\left(\tau ,\theta \right),$$
where dimensionless density and temperature are, $\theta =\rho /\rho_{c}$ and $\tau =T_{c}/T$ respectively, the subscript «c» corresponds to the critical point, the index «0» to the ideal gas state, and «e» is the excess part. Pressure, entropy, and other thermodynamic functions can be derived from the free Helmholtz energy using Maxwell’s relations. The pressure was determined by the expression:(3)$$\frac{p(\tau ,\theta )}{\rho RT}=\rho^{2}{\left.\frac{\partial f}{\partial \rho }\right|}_{T}=1+\frac{\partial \phi^{e}\left(\tau ,\theta \right)}{\partial \theta },$$
entropy:(4)$$\frac{S(\tau ,\theta )}{R}=-{\left.\frac{\partial f}{\partial T}\right|}_{p}=\tau \left(\frac{\partial \phi^{0}\left(\tau ,\theta \right)}{\partial \tau }+\frac{\partial \phi^{0}\left(\tau ,\theta \right)}{\partial \tau }\right)-\phi^{e}\left(\tau ,\theta \right)-\phi^{e}\left(\tau ,\theta \right),$$
etc.

To calculate the reference values of viscosity and thermal conductivity coefficients, the expressions were used from [41,42,43,44] and [45,46,47,48] articles have been used, respectively. For calculations of thermodynamic properties of water, Julia library [49] was developed; for other substances, Julia implementation of the CoolProp library [50] was used.

For the self-diffusion coefficient, we could not find similar high-precision expressions for a wide range of states, so the experimental data were used from [51]. That work presents tables of experimental data on the diffusion coefficient for pure substances. The data presented have been compiled, starting from the middle of the 20th century, the implication of this is that many different methods and equipment were used to obtain the data, and results of different authors may differ.

For ethanol, the experimental data on the diffusion coefficient are limited by temperature of 435 K. Therefore, a molecular dynamics simulation was used, in order to calculate the diffusion coefficient at temperatures exceeding 450 K. Molecular simulation was conducted using the GPU accelerated GROMACS package [52,53] under the NVT ensemble. Systems comprised of 4000 molecules were simulated. The Nosé–Hoover thermostat was used to maintain the constant temperature. An integration step for equations of motion was 0.5 fs. Upon reaching the thermodynamic equilibrium, the motion of molecules were calculated over 2000 ps to average the diffusion coefficient.

To describe the intermolecular interaction, the potential from the work [54] was used. A comparison of the calculated values of the diffusion coefficient with experimental data showed that at temperatures above 400 K, the deviation does not exceed 5%.

## 3. Diffusion Coefficient

Using the experimental data for diffusion coefficient, it was determined that the entire phase of the diagram can be divided into different zones. Dependence of the self-diffusion coefficient is described by a single equation in each zone. Figure 1 represents the division of the phase diagram into zones for the example of water. Analysis based on the studied substances showed that the boundary between zone II and zone III passes along the temperature $T_{sw}=T_{tr}+0.36\left(T_{c}-T_{tr}\right)$, where $T_{c}$ and $T_{tr}$ are temperatures at critical and triple points, respectively. In zone I, the diffusion coefficient is calculated by equation:(5)$$D_{I}\left(T,\rho \right)=D_{0}\left(T,\rho \right)\exp\left(a_{1}\frac{S\left(T,\rho \right)-S_{0}\left(T,\rho \right)}{k_{B}}\right),$$
where *S* and $S_{0}$ are entropy and ideal part of entropy, respectively; $a_{1}$ is a parameter; $D_{0}$ is diffusion coefficient at the same density and temperature, described by the kinetic Boltzmann equation for rarefied gases [3]:(6)$$D_{I}^{*}\left(T^{*},n^{*}\right)=\frac{3}{8\sqrt{\pi }n^{*}}\frac{\sqrt{T^{*}}}{\Omega_{1,1}\left(T^{*}\right)},$$
where $D^{*}=\frac{D}{\sigma }\sqrt{\frac{m}{\varepsilon }}$ is the dimensionless diffusion coefficient, $T^{*}=\frac{k_{B}T}{\varepsilon }$ is the dimensionless temperature, $n^{*}=n\sigma^{3}$ is the dimensionless numerical density, and *m* is the mass of the molecule. The collision integral Ω was determined by curve fitting a cubic spline for tabulated values [55].

In zone II, the diffusion coefficient is determined by the equation:(7)$$D_{II}\left(T,\rho \right)=D_{I}\left(T_{c},\rho \right)\exp\left(a_{2}\frac{S\left(T,\rho \right)-S\left(T_{c},\rho \right)}{k_{B}}\right),$$
where the subscript indicates the zone number; $\frac{S\left(T,\rho \right)-S\left(T_{c},\rho \right)}{k_{B}}$ is the dimensionless difference in entropy at the required point and at the boundary of zone I and II, corresponding to the critical temperature in this case, $a_{2}$ is a parameter. A similar equation is used for zone III:(8)$$D_{III}\left(T,\rho \right)=D_{II}\left(T_{sw},\rho \right)\exp\left(a_{3}\frac{S\left(T,\rho \right)-S\left(T_{sw},\rho \right)}{k_{B}}\right).$$

Parameters for all proposed equations were fitted by means of minimizing the deviation across the entire set of experimental data. Due to the fact that, in practice, most technological processes are conducted at a relatively low pressure, the weighting coefficient of points located near the phase equilibrium line was chosen to be five times greater than that of the remaining points. The determined parameter values are presented in Table 2.

Figure 2 shows the relative deviation of the calculated values from the experimental data for all zones of the phase diagram. Each point corresponds to experimental values [51]. Red color corresponds to the points, at which the relative deviation from the experimental data exceeds 20%. Intermediate values of relative deviation are shown by the gradation of color described in the legend. The dashed lines correspond to isobars; the pressure values in bars are indicated above the corresponding lines.

For the studied substances, the diffusion coefficient with acceptable accuracy can be described by the proposed expressions. Significant deviations are observed at high pressures. It is also seen that part of the experimental data used is not consistent with each other.

## 4. Viscosity Coefficient

Numerous works are known in the literature for the viscosity description [56]. For a rarefied gas, the solution of the Boltzmann equation can be represented as the Chapman–Enskog expression [3]:(9)$$\eta_{0}^{*}\left(T\right)=\frac{5}{16\sqrt{\pi }}\frac{\sqrt{T}}{\Omega_{2,2}\left(T^{*}\right)}$$
where $\eta_{0}^{*}=\eta \frac{\sigma^{2}}{\sqrt{m/\varepsilon }}$ is the dimensionless coefficient of viscosity.

Boundaries of the zones pass at critical temperature and at the same temperature $T_{sw}$ as for diffusion coefficient. In zone I, the viscosity coefficient is described by the expression:(10)$$\eta_{I}\left(T,\rho \right)=\eta_{0}\left(T\right)\exp\left(b_{1}\frac{S\left(T,\rho \right)-S_{0}\left(T\right)}{k_{B}}\right)$$
where $b_{1}$ is a parameter. Equation (Equation 9) was used to determine the viscosity of ideal gas $\eta_{0}$. In the second and third zones, viscosity coefficient is described by the following expressions:(11)$$\eta_{II}\left(T,\rho \right)=\eta_{I}\left(T_{c},\rho \right)\exp\left(b_{2}\frac{S\left(T,\rho \right)-S\left(T_{c},\rho \right)}{k_{B}}\right)$$
(12)$$\eta_{III}\left(T,\rho \right)=\eta_{II}\left(T_{sw},\rho \right)\exp\left(b_{3}\frac{S\left(T,\rho \right)-S\left(T_{sw},\rho \right)}{k_{B}}\right)$$
where $b_{2}$ and $b_{3}$ are the parameters, presented in Table 3. In these zones, the point on the zones boundary having the same density is used as a reference point.

It can be seen from Figure 3 that the proposed expressions allow one to calculate the viscosity coefficient with an acceptable accuracy in a wide range of states. It is possible to emphasize two zones where a deviation from the experimental data exceeds 20%. The overshoot of the viscosity coefficient is observed in the near-critical zone. The transfer coefficients (viscosity and thermal conductivity) have anomalous behavior in the zone of the critical point and the question of describing the transport properties in this zone remains open.

For alcohols, the applicability area of the proposed expression is narrow compared to water and ammonia. This may be due to an insufficiently accurate description of the intermolecular interaction. It can be seen that in zone I, the values deviate over 20% at high pressure. This affects the results in zones II and III.

## 5. Thermal Conductivity Coefficient

As demonstrated by the results of the research, for the thermal conductivity of polar substances there is no need to divide each phase diagram into zones with different equations. For the entire zone of the phase diagram, the coefficient of thermal conductivity is calculated via the expression:(13)$$\lambda \left(T,\rho \right)=\lambda_{0}\left(T\right)\frac{c_{V}\left(T,\rho \right)}{3k_{B}}\exp\left(\left(c_{0}+c_{1}T\right)\frac{S\left(T,\rho \right)-S_{0}\left(T\right)}{k_{B}}\right)$$
where $\lambda_{0}$ is the thermal conductivity of the ideal gas at the same temperature, $c_{V}$ is heat capacity at constant volume, $c_{0}$ and $c_{1}$ are the parameters, as presented in the Table 4. Since the internal degrees of freedom of the molecule play an important role in the transfer of heat, this contribution in Equation (Equation 13) was considered by the relation $c_{V}/3k_{B}$, where the heat capacity of the ideal gas state is approximately $3k_{B}$.

The thermal conductivity of the ideal gas was calculated via the Chapman–Enskog equation [3]:(14)$$\lambda_{0}^{*}\left(T\right)=\frac{75}{64\sqrt{\pi }}\frac{\sqrt{T^{*}}}{\Omega_{2,2}\left(T^{*}\right)}$$
where $\lambda^{*}=\frac{\lambda \sigma^{2}}{k_{B}}\sqrt{\frac{m}{\varepsilon }}$ is the dimensionless coefficient of thermal conductivity.

It can be seen from the Figure 4 that the proposed expression allows one to describe the coefficient of thermal conductivity with an accuracy of 20% in almost all state regions. An exception is the area of critical point. For alcohols, as in the case of the diffusion coefficient, a deviation is observed in the supercritical region as well as at high pressures. For ammonia, the intermolecular interaction potential used shows the deviation in the behavior of the thermal conductivity coefficient, which is evident from the deviation in the low rarefied gas region.

## 6. Conclusions

In this article, we proposed empirical Equations (Equation 5)–(Equation 8), (Equation 10)–(Equation 13), that relate the transport properties of polar substances to entropy. The proposed expressions allow one to calculate the diffusion, viscosity, and conductivity coefficients with an acceptable accuracy for practical use. The proposed expression has a small number of parameters, which were determined in this article using experimental data for water, ammonia, methanol, and ethanol.

## Figures and Tables

**Figure 1 entropy-22-00013-f001:**
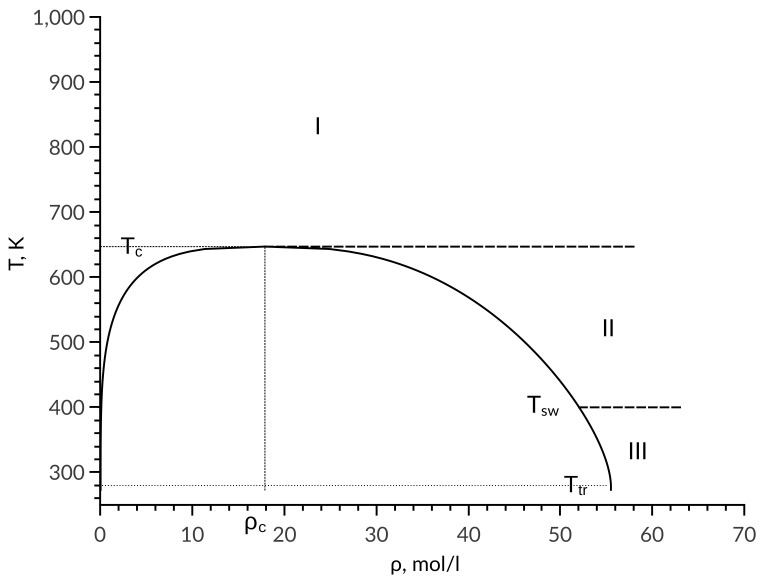
Phase diagram division into zones.

**Figure 2 entropy-22-00013-f002:**
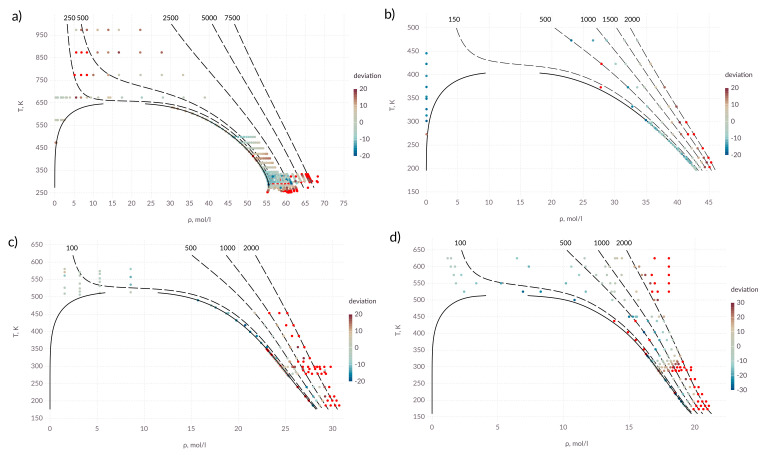
Percentage deviation in diffusion coefficient in *T*–ρ diagram; (**a**) water; (**b**) ammonia; (**c**) methanol; (**d**) ethanol; points in which the deviation is more than 20% (30% for ethanol) are shown in red; solid lines—saturation line; dashed lines—isobars, pressure values are indicated above the corresponding lines in bars.

**Figure 3 entropy-22-00013-f003:**
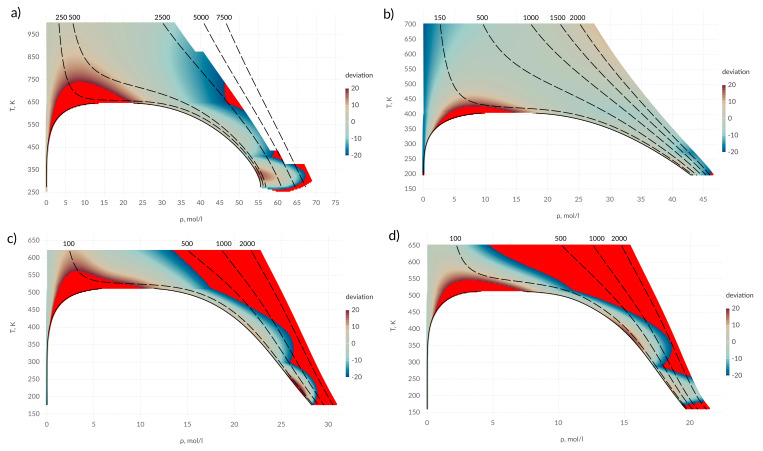
Percentage deviation in viscosity coefficient in the *T*–ρ diagram; (**a**) water; (**b**) ammonia; (**c**) methanol; (**d**) ethanol; points in which the deviation exceeds 20% are shown in red; solid lines—saturation line; dashed lines— isobars, pressure values are indicated above the corresponding lines in bars.

**Figure 4 entropy-22-00013-f004:**
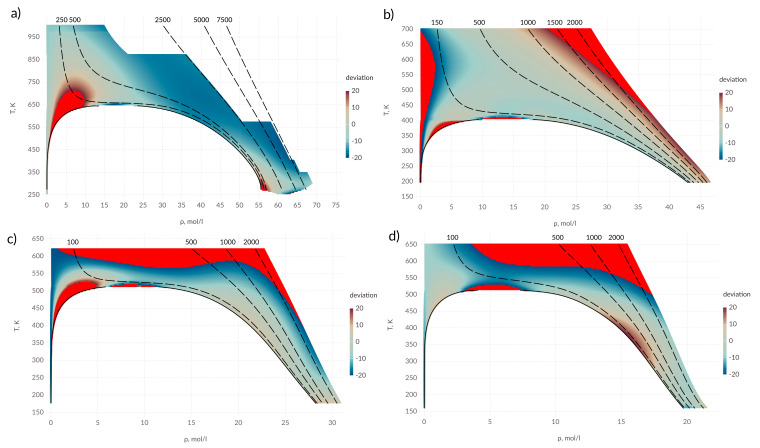
Percentage deviation in thermal conductivity coefficient in the *T*–ρ diagram; (**a**) water; (**b**) ammonia; (**c**) methanol; (**d**) ethanol; points in which the deviation exceeds 20% are shown in red; solid lines—saturation line; dashed lines—isobars, pressure is indicated above the lines in bars.

**Table 1 entropy-22-00013-t001:** Parameters of the Stockmayer potential for the substances.

Substance	$\varepsilon /k_{B}$, K	σ, nm	δ
water [35]	470	0.259	1.5
ammonia [36]	309.9	0.3215	1.52
ethanol [36]	385.2	0.3657	1.1
methanol [36]	373.3	0.4299	0.855

**Table 2 entropy-22-00013-t002:** Parameters for the calculation of the diffusion coefficient.

Substance	$a_{1}$	$a_{2}$	$a_{3}$
water	0.0	0.2637	0.7621
ammonia	0.0388	0.3707	0.6264
methanol	0.03364	0.4212	0.8633
ethanol	0.1748	0.2804	0.8387

**Table 3 entropy-22-00013-t003:** Parameters for calculation of viscosity coefficient.

Substance	$b_{1}$	$b_{2}$	$b_{3}$
water	−0.4019	−0.1914	−0.6052
ammonia	−0.6809	−0.151	−0.3485
methanol	−0.4132	−0.2940	−0.7253
ethanol	−0.3881	−0.2665	−0.6644

**Table 4 entropy-22-00013-t004:** Parameters for calculation of thermal conductivity coefficient.

Substance	$c_{1}$	$c_{2}$
water	−0.1803	−0.0006337
ammonia	−0.3915	−0.001032
methanol	−0.344	0.0001427
ethanol	−0.2914	0.0001376

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
