# Peer review of "Relationship between the Transport Coefficients of Polar Substances and Entropy"

_entropy, 2019, doi:10.3390/e22010013_

Round 1
Reviewer 1 Report
The paper is an excellent contribution to description of rheological properties of polar substances. It is based on a proposed empirical expression relating the transport properties to the entropy. The proposed expression has a small number of parameters, which can be determined using experimental data. Authors have used this approach to calculate the diffusion, viscosity and conductivity coefficients with an acceptable accuracy for practical use of several polar materials – water, ammonia, methanol and ethanol. Data obtained by authors show a good agreement with experimental data with limited deviation observed only in the region near the critical point and at high pressures.
The paper is of good quality and is recommended to publication. Authors are however advised to note that theoretical expressions relating viscosity (and diffusion coefficients) to entropy are well known and used (!) in describing behaviour of amorphous materials – see for example the recent publication that gives explicit equations very much analogous to expressions used by authors: M.I. Ojovan. Thermodynamic Parameters of Bonds in Glassy Materials from Shear Viscosity Coefficient Data. International Journal of Applied Glass Science, 5, (1) 22–25 (2014), DOI: 10.1111/ijag.12045
Author Response
Thank you for the review of the article and these comments.
Point 1
The paper is of good quality and is recommended to publication. Authors are however advised to note that theoretical expressions relating viscosity (and diffusion coefficients) to entropy are well known and used (!) in describing behaviour of amorphous materials – see for example the recent publication that gives explicit equations very much analogous to expressions used by authors: M.I. Ojovan. Thermodynamic Parameters of Bonds in Glassy Materials from Shear Viscosity Coefficient Data. International Journal of Applied Glass Science, 5, (1) 22–25 (2014), DOI: 10.1111/ijag.12045
Responce 1
We read this article and added it to a brief overview of the methods used in the introduction to the article.
Reviewer 2 Report
The manuscript "Relationship between the Transport coefficients of polar substances and the entropy" falls within the scope of the Journal. It is interesting from a theoretical Point of view. It can be seen that the develoment of the equations has been worked for longer. However, there are some Points that can be arranged before the manuscript can be published, namely:
It is suggested that a native english speaker Review the wording and editing of the article. There are so many dummy citations, it would be better to avoid as much as possible this practice or, explain why they are worth being mentioned in the manuscript. It is suggested to provide a number of Equation to the expression for the pressure. Page 3. It is suggested to provide the range of temperature, pressure or any other parameter within which the proposed equations are valid. To provide in the conclusions which are the disadvantages of using the proposed equationsAuthor Response
The authors are grateful to the reviewers for the careful review of the article and the valuable comments. Based on the comments made, the following changes were made:
Point 1
It is suggested that a native english speaker Review the wording and editing of the article.
Responce 1
We made changes to the text of the article.
Point 2
There are so many dummy citations, it would be better to avoid as much as possible this practice or, explain why they are worth being mentioned in the manuscript.
Responce 2
Errors were made in the links to sources in Russian (ref. 13-15 and). We swapped surnames and first names in biblatex files. Also, the URLs of those sites on the Internet from where the information was taken were not indicated in the links. The remaining links look quite reasonable.
We also correct the last paragraph of the introduction, we hope that it has become more understandable.
Point 3
It is suggested to provide a number of Equation to the expression for the pressure.
Responce 3
We presented expressions on separate lines.
Point 4
Page 3. It is suggested to provide the range of temperature, pressure or any other parameter within which the proposed equations are valid. To provide in the conclusions which are the disadvantages of using the proposed equations.
Responce 4
It is rather difficult to say the exact boundaries of the region in which the proposed expressions work, since the magnitude of this region for each substance depends on temperature. We correct figures 3-4 adding isobar lines to them. It seems to us that this will allow to understand the scope of applicability of the proposed expressions.
Reviewer 3 Report
This manuscript investigated several physical properties of polar substances. The diffusion coefficient, viscosity coefficient, and thermal conductivity coefficient were calculated by the proposed expressions. They showed good agreement as compared with experimental data. The results of this manuscript are useful to obtain the reference values of some transport properties. The conclusions are supported by the data. This manuscript is acceptable after revising according to the following comments.
The sources of the proposed empirical expressions relating to transport properties of polar substances, such as Eqs. 1-11, should be clearly indicated.
Please show the sources of parameters’ values in Tables 2-4.
Please isolate equations in the following line, but not in the sentences, e.g. lines 82 and 106.
Please indicate the meaning of dash line in Figs. 2-4.
The expression “(??)” in Line 140 should be corrected.
Author Response
Thank you for the review of the article and these comments.
Point 1
The sources of the proposed empirical expressions relating to transport properties of polar substances, such as Eqs. 1-11, should be clearly indicated.
Responce 1
We have provided links to the expressions used. We also indicated the numbers of the proposed expressions in the conclusion.
Point 2
Please show the sources of parameters’ values in Tables 2-4.
Responce 2
These parameter values were adjusted according to experimental data or according to high-precision approximations with many parameters. We made correction to the text of the article
Point 3
Please isolate equations in the following line, but not in the sentences, e.g. lines 82 and 106.
Responce 3
Fixed
Point 4
Please indicate the meaning of dash line in Figs. 2-4.
Responce 4
We correct these drawings and hope that they become more understandable. In the new version, isobars are indicated by dashed lines.
Point 5
The expression “(??)” in Line 140 should be corrected.
Responce 5
Fixed